# Field Evaluation of the Dust Impacts from Construction Sites on Surrounding Areas: A City Case Study in China

**Hui Yan [1], Guoliang Ding [1], Hongyang Li [1], Yousong Wang [1], Lei Zhang [2], Qiping Shen [3] and Kailun Feng [4,*]**

[1] Department of Construction Management, South China University of Technology, Guangzhou 510641, China; cthyan@scut.edu.cn (H.Y.); ctdingguoliang@mail.scut.edu.cn (G.D.); cthyli@scut.edu.cn (H.L.); yswang@scut.edu.cn (Y.W.)

[2] Department of Construction Management, Guangzhou University, Guangzhou 510006, China; somzhanglei@gzhu.edu.cn

[3] Department of Building and Real Estate, The Hong Kong Polytechnic University, Hong Kong 999077, China; geoffrey.shen@polyu.edu.hk

[4] Department of Construction Management, Harbin Institute of Technology, Harbin 150001, China

[*] Correspondence: kailunfeng@hit.edu.cn; Tel.: +86-0451-8640-2180

**Abstract:** Construction activities generate a large amount of dust and cause significant impacts on air quality of surrounding areas. Thus, revealing the characteristics of construction dust is crucial for finding the way of reducing its effects. To fully uncover the characteristics of construction dust affecting surrounding areas, this study selected seven representative construction sites in Qingyuan city, China as empirical cases for field evaluation. In the experiment, the up-downwind method was adopted to monitor and collect TSP (total suspended particulate), $PM_{10}$ and $PM_{2.5}$ (particulate matter $\leq$10 μm and 2.5 μm in aerodynamic diameter, respectively) concentrations, meteorological data and construction activities of each site for 2 to 3 days and 18 h in a day. The results show that the average daily construction site makes the surrounding areas' concentration of TSP, $PM_{10}$ and $PM_{2.5}$ increase by 42.24%, 19.76% and 16.27%, respectively. The proportion of TSP, $PM_{10}$ and $PM_{2.5}$ in building construction dust is 1, 0.239 and 0.116, respectively. The large diameter particulate matter was the major constituent and the distance of its influence was limited. In addition, construction vehicles were one of the main influencing factors for building construction dust. However, building construction dust was not significantly correlated with any single meteorological factor when it did not change too much. Findings of this research can provide a valuable basis for reducing the impact of building construction dust on surrounding areas.

**Keywords:** construction dust; TSP; $PM_{10}$; $PM_{2.5}$; surrounding areas; impact characteristics

## 1. Introduction

Atmospheric pollution has become an increasingly serious problem in some urban areas especially where experiencing fast developing and urbanization. Particulate matter is a primary pollutant in all of atmospheric air pollution. Through analyzing of the source of particulate matter it was found that the dust is one of the main sources contributing atmospheric particulate matter pollution in cities [1–4] and that building construction is a major source leading to city dust [5,6].

Building construction dust refers to the diffusion of particulate matter in the atmosphere caused by a construction site and activities—which has a great negative impact on human health and influencing the surrounding areas of the construction site [7–11]. In order to find ways of reducing these negative

effects, many scholars have performed research on building construction dust, mainly focusing on the emission characteristics [12,13], diffusion law [14,15], emission calculation methods [16–19] and the main influencing factors [20,21]; some valuable results have been revealed for impact reduction.

The indicators to measure the building construction dust includes dust fall, TSP (total suspended particulate), $PM_{10}$ and $PM_{2.5}$ (particulate matter $\leq 10$ μm and 2.5 μm in aerodynamic diameter, respectively) [10,20,22,23]. Most previous research only adopted one or two monitoring indicators, which cannot contain all emissions' impact characteristics. More importantly, most studies [21,24] arranged monitoring points inside the construction site, which can only reflect the pollution of building construction dust inside the construction area, but cannot reflect the pollution on the surrounding area outside the construction boundary.

In order to study the pollution characteristics and main influencing factors of building construction dust on the surrounding areas, this paper used the up-downwind direction method [25], which treats the construction site as an unorganized emission source and places the monitoring points at the boundary of the up and down wind direction of the construction site. TSP, $PM_{10}$ and $PM_{2.5}$ were selected as monitoring indicators that reflects the emission of particulate matter with different particle sizes. This monitoring plan was applied to seven construction sites in Qingyuan City. The characteristics of the impact of building construction dust on surrounding areas and the main factors influencing impacts were studied to provide a theoretical basis for the control of building construction dust.

The paper is organized as follows. It begins with a literature review, which introduces the research status of the characteristics of emission and main influencing factors of the building construction dust, and the limitations of the current research are summarized. Section 3 details the monitoring methods for this study. Section 4 mainly analyzes the monitoring results of seven construction sites in Qingyuan City, including the analysis of the overall situation, the impact of building construction dust on the surrounding environment, the distribution of particulate matter size, etc. In Section 5, statistical analysis is used to identify and analyze the main factors affecting building construction dust from the external environment and construction intensity. The concluding section summarizes the theoretical and practical contributions of this research, as well as future research perspectives.

## 2. Literature Review

In order to better accomplish the research objectives of this paper, it is necessary to understand the state-of-the-art and limitations of previous construction dust research. In this section, existing studies will be reviewed and analyzed from the perspectives of dust evaluation indicators, emission characteristics and main factors influencing building construction dust.

### 2.1. Evaluation Indicators and Emission Characteristics of Building Construction Dust

Tian et al. [14,26] proved that the dust fall (DF) indicator has a good correlation with TSP and $PM_{10}$ in vertical and horizontal diffusion laws; therefore, early studies mostly adopted dust fall as a monitoring indicator. For example, Huang et al. [27] carried out dust fall monitoring on more than 40 construction sites in the suburbs of Beijing. The dust pollution laws of different construction stages were studied. It was found that the relationship between dust pollution in different construction stages was significant. In these studies, the dust fall was used as a monitoring indicator to describe the emission characteristics of building construction dust. However, the sampling frequency of dust fall is once a month, which represents the average level of pollution within one month, and it cannot reflect the timeliness of dust pollution in construction. In addition, Muleski et al. [18] employed $PM_{10}$ and $PM_{2.5}$ emissions to conduct on-the-spot measurements from construction activities. Fan et al. [28] and Yang [29] used the $PM_{10}$ indicator to monitor the building construction dust.

Some researchers have also conducted monitoring and research on certain construction stages or construction activities. For example, Li et al. [30–32] and Huang et al. [24] used a TSP indicator to separately assess impacts of construction activities in different construction stages and found

that the cement processing areas, the woodworking shed, the sides of the road and the putty area had serious dust pollutions. Faber et al. [33] monitored the earthwork of a construction site in Germany and found that the $PM_{10}$ emissions from earthwork activities reached 44% of the total $PM_{10}$ emissions from German construction activities. Azarmi and Kumar [34] monitored the $PM_{10}$ and $PM_{2.5}$ emissions during the demolition phase. The results showed that the release of coarse particulate matter was much greater during the demolition process, and the PM in the downwind direction decreased logarithmically with distance. Azarmi et al. [10] confirmed that a large amount of particulate matter was generated during the renovation of the building. However, the monitoring methods used by various scholars to measure dust during construction are not consistent, and most studies only select one or two of them (mainly size below 10 μm) to characterize the building construction dust, which cannot completely describe the characteristics of all particulate matter generated by construction activities.

## 2.2. Main Influencing Factors of Building Construction Dust

During the construction process, the site conditions are complex and dust emissions are influenced by multiple factors. The external environment of the construction site can be an important source of factors. Araújo et al. [21] found that the weather condition has an important influence on the concentration of particulate matter during the on-site monitoring of the particulate matter at the construction site. However, due to the limited information about the measurement concentration, they failed to correlate the weather condition variables with PM concentration. Furthermore, Zhang et al. [35] found that the building construction dust emission has obvious seasonal changes, which is consistent with the research results of Zhao et al. [36]. Luo [13] further studied the relationship between building construction dust and meteorological factors. It was found that building construction dust was significantly positively correlated with wind speed and relative humidity, and weakly correlated with temperature. In addition, dust moisture content is also one of the influencing factors [37]. The following is the internal activities of construction sites, which are also one of the main influencing factors of building construction dust. Kinsey et al. [38] found that vehicles driving out of the construction site can carry a large amount of dust and sediment to nearby roads, causing secondary dust to rise under external force. Azarmi et al. [39] performed detailed monitoring on concrete mixing, drilling and cutting activities. Peaks of particulate matter (including $PM_{10}$, $PM_{2.5}$ and $PM_{0.1}$) during drilling and cutting activities were four and 14 times higher than the background value. Moraes et al. [40] focused on monitoring the concentration of particulate matter ($PM_{10}$) produced by concrete and masonry in construction activities. All these studies indicated that construction activities are an important influencing factor of building construction dust [28]. A comprehensive analysis of the above research results shows that the main influencing factors of building construction dust can be attributed to two categories: environmental factors and construction activity factors. This also provides a basis for the development of the experimental monitoring and factorial analysis for this study.

The above-mentioned research on building construction dust is mainly reflected the dust pollution inside the construction site, e.g., Muleski et al. [18], Araújo et al. [21] and Moraes et al. [40], whose monitoring points were all placed inside the construction site or near some construction activities. Although Azarmi et al. [41] evaluated the impact of $PM_{10}$ and $PM_{2.5}$ generated by construction activities on the surrounding area of the site, the data source for the study was the monitoring station around the site, and not all changes in particulate matter were caused by building construction. In addition, the concentration of particulate matter in that study was taken an annual average value and cannot reflect the short-term effects of construction activities.

Based on the above analysis, we can clearly find that there are two limitations of present construction dust researches. Firstly, the selection of monitoring indicators is simplified, which cannot fully reflect the pollution of building construction dust. Secondly, the monitoring points are mostly inside the construction site, thus, it is impossible to describe the dust pollution caused by construction to the nearby outside of the construction site. Based on the existing research, this paper

will conduct further empirical case study and analysis on the above two limitations, and hope to supplement the current knowledge in building construction dust field.

## 3. Methods

This study used on-site monitoring nearby the construction sites to collect data. The monitoring plan included three aspects; monitoring indicators, monitoring methods and monitoring samples, which will be described in detail in this section. Then, the monitoring data was processed to analyze the monitoring results from multiple angles. Emission characteristics and major influencing factors can be revealed based on the results of data processing. The procedures of the conducted research are shown in Figure 1.

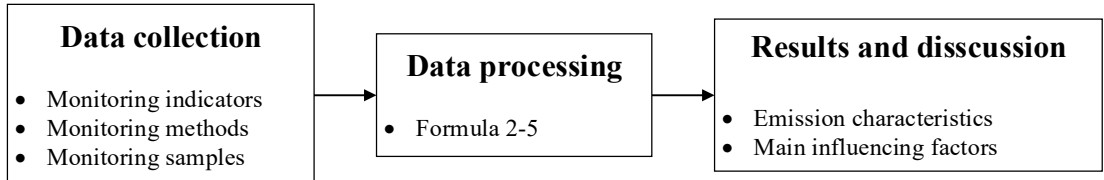

**Figure 1.** Research framework.

### *3.1. Monitoring Indicators*

There are four main indicators for measuring building construction dust: dust fall, TSP, $PM_{10}$ and $PM_{2.5}$ [10,20,22,23]. The advantage of dust fall as a monitoring indicator is the low monitoring cost and easy operations. However, the sampling frequency is once a month, which represents the dust pollution during a month; thus, this cannot reflect the real-time dust pollution. Therefore, this indicator was not selected in this study. TSP, $PM_{10}$ and $PM_{2.5}$ were sampled once a day, reflecting the timeliness of 24 h average pollution situation. Currently, TSP, $PM_{10}$ and $PM_{2.5}$ are important indicators for measuring air quality standards. In the study of air quality in the Xinjiang Province, it was found that using different monitoring indicators such as $PM_{10}$ and TSP have a greater impact on the evaluation of air quality level [42]. Therefore, this study selects TSP, $PM_{10}$ and $PM_{2.5}$ as monitoring indicators, which, respectively, represent particles with aerodynamic diameters less than 100 μm, 10 μm and 2.5 μm. They contain different particle sizes of building construction dust, and can more comprehensively reflect the impact of building dust on air quality.

According to the requirements of China's "Environmental Air Quality Standards" [43], the seven sites monitored in this study are located in the second ambient air functional zone, which is applicable to level II concentration limit (Table 1). The monitoring of three ambient air pollutants of TSP, $PM_{10}$ and $PM_{2.5}$ was a 24 h average.

**Table 1.** $PM_{10}$ particle concentration limit table.

| Monitoring Indicator | Ambient Air Functional Area Level | Unit | Concentration Limit | Reference |
|---|---|---|---|---|
| TSP | level II | $μg/m^3$ | 300 | |
| $PM_{10}$ | level II | $μg/m^3$ | 150 | [43] |
| $PM_{2.5}$ | level II | $μg/m^3$ | 75 | |

### *3.2. Monitoring Methods*

#### 3.2.1. Dust Concentration Monitoring

In this study, three large-flow air particulate samplers were set up at each sampling point that separately sampled TSP, $PM_{10}$ and $PM_{2.5}$. The instrument model was air/smart integrated sampler (2050) and the monitoring height was 1.5 m. The $PM_{10}$ and $PM_{2.5}$ collection methods and equipment

meet the requirements of the "Determination of atmospheric articles $PM_{10}$ and $PM_{2.5}$ in ambient air by gravimetric method" [44]. The TSP collection method and equipment satisfy the requirements of the "Ambient air-Determination of total suspended particulates-Gravimetric method" [45]. Sample analysis and result calculation were performed after sampling. The concentrations of TSP, $PM_{10}$ and $PM_{2.5}$ were calculated according to Formula (1):

$$C = \frac{C_a - C_b}{V} \tag{1}$$

where C is average concentration of particulate matter at the time of sampling ($\mu g/m^3$); $C_a$: Membrane quality after sampling ($\mu g$); $C_b$: Membrane quality before sampling ($\mu g$); V: Sampling volume converted to standard state (101.325 kPa, 273 K) ($m^3$).

Unorganized emissions are the irregular discharge of atmospheric pollutants without passing through the exhaust funnel, and the construction site is a typical unorganized emission source. Unorganized emission monitoring point setting method in the "Integrated Emission Standard of Air Pollutants" [25] is an important reference for the monitoring point setting method of this study. A reference point was set on the upwind direction of the construction site located at the center of the boundary of the construction site and within a sector of 2 m to 50 m from the boundary. 2–4 monitoring points were set within 10 m outside the boundary of the downwind direction, as shown in Figure 2. The specific layout location can be further referred at the "Technical Guidelines for Fugitive Emission Monitoring of Air Pollutants" [46], taking into account of factors such as meteorological conditions (wind direction, wind speed, illumination, etc.), building distribution, site entrances and exits, site boundaries, etc. It cannot be set on the construction site entrances and exits and the sides of construction road. In order to avoid the contingency of monitoring data, each construction site was continuously monitored for 2–3 days, and the dust concentration data of the up-down wind direction of the construction site can be accordingly obtained.

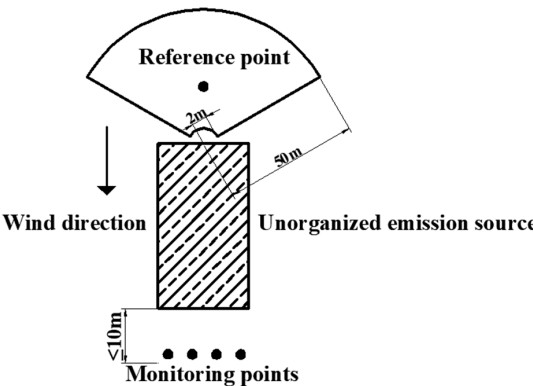

**Figure 2.** Schematic diagram of setting monitoring points.

In this study, according to the monitoring setting of unorganized emission sources, Formulas (2)–(5) were used to conduct quantitative analysis of building construction dust emitted from seven monitoring sites. Up-down wind direction incremental concentration (incremental concentration for short) reveals the absolute value of incremental concentration of the building construction dust (Formula (2)). The relative incremental concentration indicates the percentage of the downwind direction concentration rising compared with the upwind direction concentration (Formula (3)). The average of the incremental concentration and the average of the relative incremental concentration are calculated through Formulas (4) and (5):

$$\text{Incremental concentration} = Max\{\text{Downwind direction concentration}\} - \text{Upwind direction concentration} \tag{2}$$

$$\text{Relative incremental concentration} = \frac{\text{Incremental concentration}}{\text{Upwind direction concentration}} \times 100\% \tag{3}$$

$$\text{Average of incremental concentration} = \frac{\sum \text{Incremental concentration}}{\text{Monitoring days}} \tag{4}$$

$$\text{Average of relative incremental concentration} = \frac{\sum \text{Relative incremental concentration}}{\text{Monitoring days}} \tag{5}$$

### 3.2.2. Meteorological Data Monitoring

The rainy weather has an obvious inhibitory effect on the dust [47] and the monitoring equipment is not appropriate for working in rainy weather. Therefore, sunny and cloudy days were selected as monitoring time in this study. On the construction site, wind direction anemometer, temperature hygrometer and air pressure box were used to obtain meteorological data such as wind speed, temperature, humidity and atmospheric pressure. The average value of a day was used for data analysis.

### 3.2.3. Construction Activity Data Collection

By considering the availability of data and the feasibility of lateral comparison between seven construction sites, this study quantified construction activities by construction intensity. Daily working hours (DWH) and daily transport trips (DTT) were used to characterize the construction intensity. DWH are the working hours multiply the number of workers, and DTT is the number of construction vehicles entering and leaving the construction site.

### 3.3. Selection of the Monitoring Samples

Qingyuan city, located in the central part of the Guangdong Province in southern China, is a rapidly developing city. In recent years, with the acceleration of urbanization, the number of construction sites in the city has increased rapidly. This might cause increased air pollution in Qingyuan city, especially the areas nearby construction site. In addition, Qingyuan City pays more attention to urban air quality problems because of the creation of a civilized city. Thus, it is particularly important to measure the impact of building construction activities on the surrounding places.

The monitoring construction sites selected are all located in Qingyuan. Seven representative sites were selected as monitoring samples in this study, considering factors of the leading wind direction, the location of the environmental monitoring point, the construction stage of the project, the scale of the project, the location and concern of the construction site. The selected seven construction sites contain different stages of construction and thus can represent the average level of dust emissions. Table 2 shows the specific information of the monitoring construction sites. Figure 3 shows the plan and monitoring setting of construction sites. Pentagrams indicate the location of the monitoring point and the arrow indicates the leading wind direction.

**Table 2.** Information of the monitoring construction sites.

| NO. | ProjectIndex | Number of Stores (Aboveground/Underground) | Structure Type | Usage | Construction site Area (m²) | Construction Stage * |
|-----|-------------|--------------------------------------------|----------------|-------|------------------------------|----------------------|
| 1 | A | 18/1 | Frame-shear wall structure | Apartment | 33,500 m² | Foundation and main engineering |
| 2 | B | 22/2 | Frame-shear wall structure | Commercial and residential | 7087.63 m² | Foundation, main and decoration engineering |
| 3 | C | 31/2 | Frame-shear wall structure | Residential | 35,511.07 m² | Main engineering |

**Table 2.** *Cont*.

| NO. | ProjectIndex | Number of Stores (Aboveground/Underground) | Structure Type | Usage | Construction site Area (m²) | Construction Stage * |
|---|---|---|---|---|---|---|
| 4 | D | 31/2 | Frame-shear wall structure | Commercial and residential | 32,504.66 m² | Decoration engineering |
| 5 | E | 31(32)/2 | Frame-shear wall structure | Commercial and residential | 39,823.07 m² | Foundation, main and decoration engineering |
| 6 | F | 32/1 | Frame-shear wall structure | Residential | 51,589.97 m² | Main engineering |
| 7 | G | 32/1 | Frame-shear wall structure | Commercial and residential | 42,362.92 m² | Foundation and decoration engineering |

* Note: Organize according to the leading construction phase at the construction sites, each construction site could contain multiple construction phases.

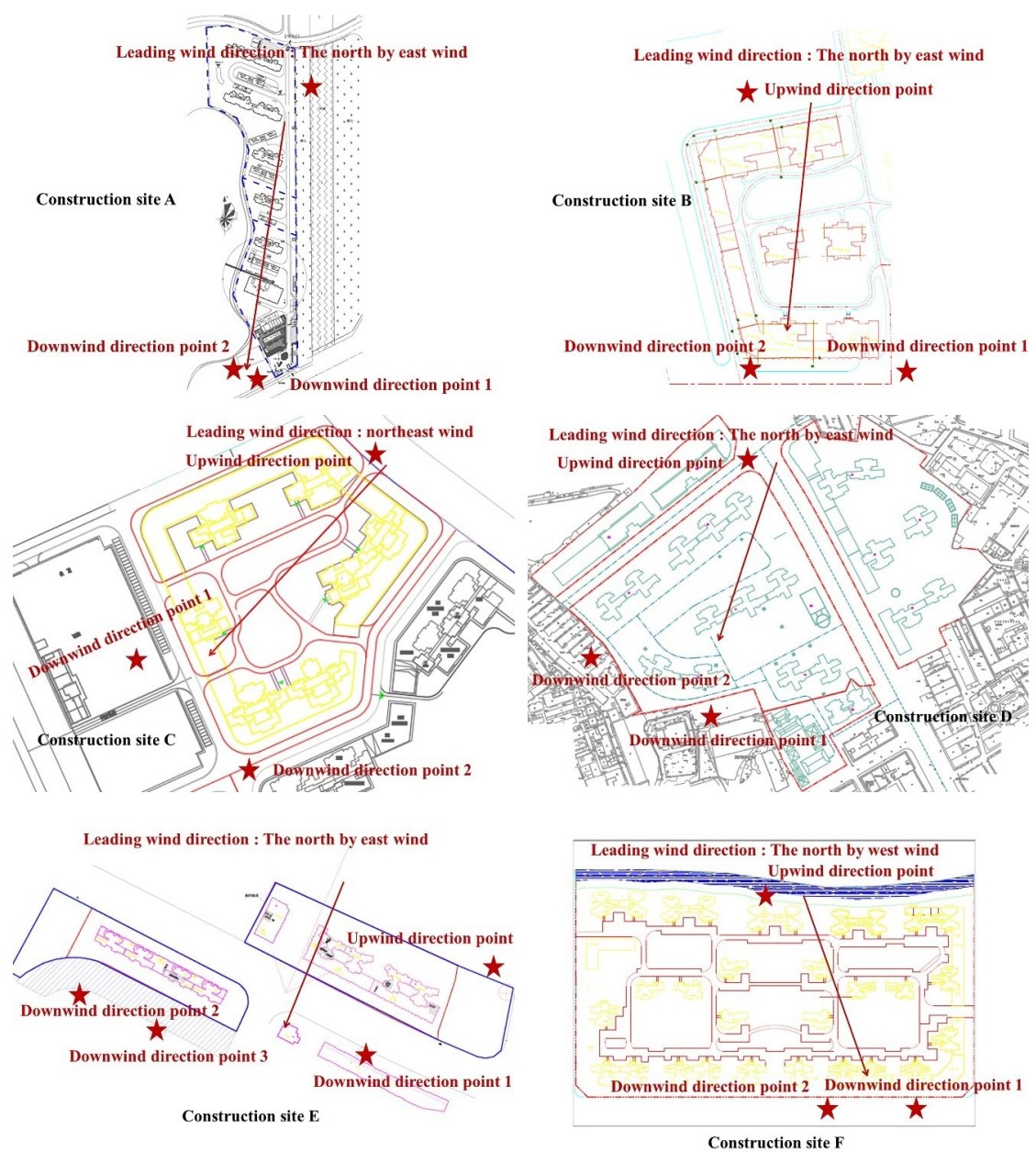

**Figure 3.** *Cont*.

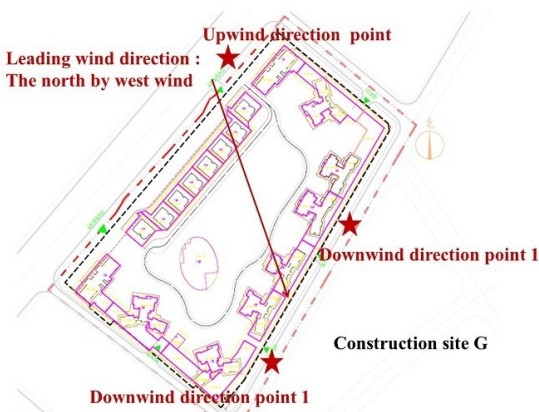

**Figure 3.** Plans of construction sites monitoring points setting.

## 4. Results and Discussion

According to the above experimental method, on-site monitoring was organized on the seven construction sites. The monitoring results are summarized in Figure 4. This figure shows the location, monitoring date and monitoring results of each monitoring site. The monitoring was conducted between November 2017 and March 2018, when the Qingyuan was in the winter and spring, respectively. The first column is the upwind direction concentration of each indicator. The second column is the maximum downwind direction concentration of each indicator. The third column is the official concentration of the nearby environmental monitoring station, which represents the basic ambient air quality of the day. The TSP concentration is not counted by the environmental monitoring station in Qingyuan. The darker the color is, the higher the concentration value is. It can be clearly seen that from the absolute value of air quality, Site A had the best air quality on the day of monitoring, while Site G had the worst air quality on the day of monitoring. However, the absolute value of air quality does not reflect the dust pollution caused by construction, which will be properly assessed by incremental indicators. SPSS 23.0 statistical software was used for data analysis in this study.

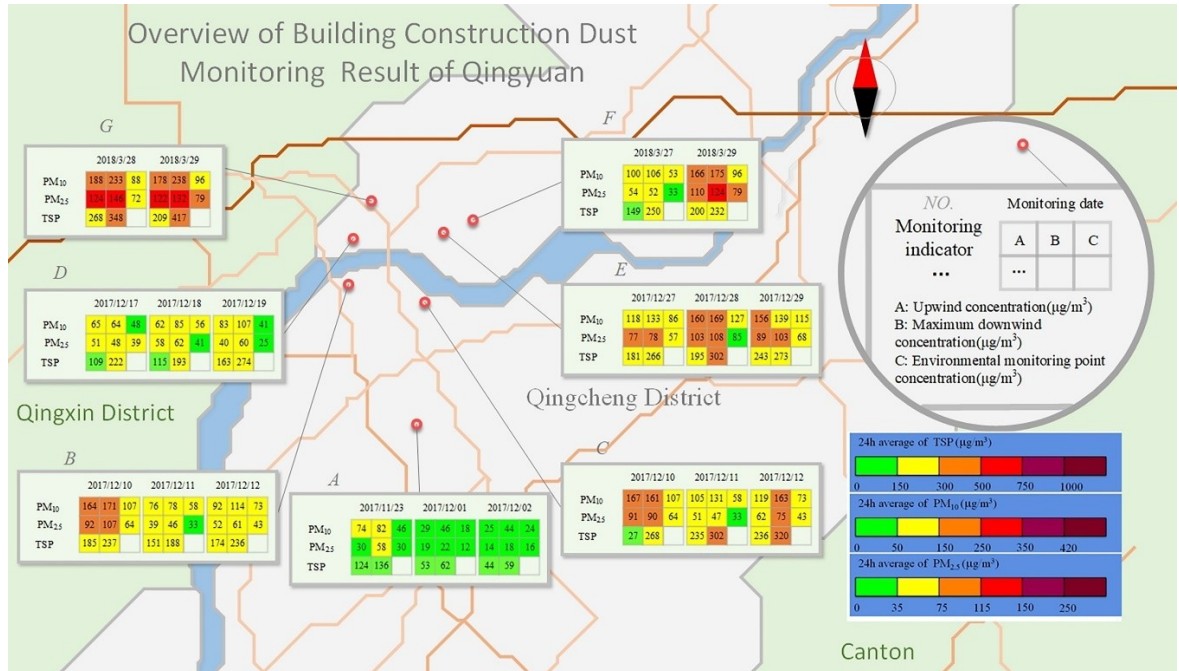

**Figure 4.** Overview of building construction dust monitoring results of Qingyuan.

*4.1. Analysis of Overall Situation*

The compliance of monitoring data according to the limit value of particulate matter concentration is shown in Table 3.

**Table 3.** The compliance of monitoring indicators.

| Indicators | Total Number of Monitoring Points | The Number of Monitoring Points Not Up to Local Standard | Over-Standard Rate | Up-to-Standard Rate |
|---|---|---|---|---|
| TSP | 60 | 6 | 10.00% | 90.00% |
| $PM_{10}$ | 60 | 19 | 31.67% | 68.33% |
| $PM_{2.5}$ | 60 | 25 | 41.67% | 58.33% |

It can be found from Table 3 that there is a total of 60 TSP up-down wind direction monitoring data, of which six monitoring points exceeded the concentration limit of 300 $\mu g/m^3$. The over-standard rate was 10%; there was a total of 60 $PM_{10}$ up-down wind monitoring data. Among them, 19 monitoring data exceeded the concentration limit of 150 $\mu g/m^3$. The over-standard rate was 31.67%; there was a total of 60 $PM_{2.5}$ up-down wind monitoring data, of which 25 points exceeded the concentration limit of 75 $\mu g/m^3$. The over-standard rate was 41.67%.

Combined with Figure 3, the specific analysis of each over-standard point data shows that for $PM_{10}$ and $PM_{2.5}$, the up-down downwind direction concentration exceeded the standard at the same time, indicating that the $PM_{10}$ and $PM_{2.5}$ concentration in the basic ambient air had exceeded the standard. While for TSP, the downwind direction concentration exceeded the standard, which indicates that the construction site was the direct cause of the TSP concentration exceeding the standard.

From other points of view, for the seven sites monitored, the overall up-to-standard rate of different monitoring indicators had a wide gap, with TSP up-to-standard rate as the highest, 90%, followed by $PM_{10}$, 68.33%, and $PM_{2.5}$ as the lowest, only 58.33%. This shows that if different monitoring indicators are used to evaluate the surrounding environmental quality of the construction site, different results will be obtained. Each monitoring indicator reflected one aspect of building construction dust pollution. The results above further verified the rationality and comprehensiveness of the selection of monitoring indicators in this study.

*4.2. Analysis of Up-Downwind Direction Concentration*

In order to analyze the change of up-downwind direction concentration after passing through the construction site, this section uses a paired sample *T*-test statistical method to conduct paired analysis on 19 sets of TSP, $PM_{10}$ and $PM_{2.5}$ data of up-downwind direction concentration. The paired sample *T*-test was to compare whether there was a significant difference in the mean values of the paired up-downwind direction concentrations of each monitoring indicator. The null hypothesis was that there was no significant difference in the mean of the up-downwind direction concentration. The premise of using the paired sample *T*-test was that the independent sample variables satisfy the normality.

Firstly, the normality test was performed on 19 sets of data. The significance of the TSP, $PM_{10}$ and $PM_{2.5}$ up-downwind concentrations were all greater than 0.05 (Table 4), which indicates that these data all obeyed the normal distribution. After the premise hypothesis verification, the paired sample *T*-test was performed on the up-downwind direction data. The results are shown in Tables 5 and 6.

**Table 4.** Tests of normality.

| | Kolmogorov-Smirnov [a] | | |
| --- | --- | --- | --- |
| | **Statistic** | **df (Degree of Freedom)** | **Significance** |
| Upwind direction TSP concentration | 0.086 | 19 | 0.200 * |
| Downwind direction TSP concentration | 0.144 | 19 | 0.200 * |
| Upwind direction $PM_{10}$ concentration | 0.176 | 19 | 0.126 |
| Downwind direction $PM_{10}$ concentration | 0.098 | 19 | 0.200 * |
| Upwind direction $PM_{2.5}$ concentration | 0.141 | 19 | 0.200 * |
| Downwind direction $PM_{2.5}$ concentration | 0.173 | 19 | 0.137 |

* This is a lower bound of the true significance. [a] Lilliefors Significance Correction.

**Table 5.** Paired samples test.

| | | **t** | **df** | **Significance (Two-Tailed)** |
| --- | --- | --- | --- | --- |
| Pair 1 | Upwind direction TSP concentration-Downwind direction TSP concentration | −6.540 | 18 | 0.000 |
| Pair 2 | Upwind direction $PM_{10}$ concentration-Downwind direction $PM_{10}$ concentration | −3.851 | 18 | 0.001 |
| Pair 3 | Upwind direction $PM_{2.5}$ concentration-Downwind direction $PM_{2.5}$ concentration | −4.064 | 18 | 0.001 |

**Table 6.** Paired samples correlations.

| | | **N** | **Correlation** | **Significance** |
| --- | --- | --- | --- | --- |
| Pair 1 | Upwind direction TSP concentration & Downwind direction TSP concentration | 19 | 0.866 | 0.000 |
| Pair 2 | Upwind direction $PM_{10}$ concentration & Downwind direction $PM_{10}$ concentration | 19 | 0.945 | 0.000 |
| Pair 3 | Upwind direction $PM_{2.5}$ concentration & Downwind direction $PM_{2.5}$ concentration | 19 | 0.970 | 0.000 |

It can be seen from Tables 5 and 6 that the significance of the TSP, $PM_{10}$ and $PM_{2.5}$ concentration paired samples in the up-downwind direction is less than 0.05, which indicate the null hypothesis is rejected. It shows that the data of the up-downwind direction concentration has a significant difference, which reflects that the construction site has a significant impact on the concentration of particulate matter. It causes the downwind direction concentration of particulate matter to increase significantly. In addition, the correlation coefficients of TSP, $PM_{10}$ and $PM_{2.5}$ concentrations of the up-downwind direction concentration are 0.866, 0.945 and 0.970, respectively. Although the up-downwind direction concentration had strong correlation, there were still some differences. TSP up-downwind direction concentration had the weakest correlation, $PM_{10}$ correlation was second, and $PM_{2.5}$ had the least correlation. This shows that the construction site has the greatest impact on the TSP concentration in the air, followed by the $PM_{10}$ concentration and the $PM_{2.5}$ concentration.

*4.3. Analysis of Up-Downwind Direction Incremental Concentration*

By using the above Formulas (2)–(5), the calculation results of the up-down wind direction incremental concentration of the seven construction sites are shown in Table 7. The increments reflect the influence of building construction dust on the surrounding air quality.

**Table 7.** Analysis table of up-downwind direction incremental concentration of the seven construction sites.

| Construction Site | Monitoring Date | TSP Incremental Concentration | PM$_{10}$ Incremental Concentration | PM$_{2.5}$ Incremental Concentration |
|---|---|---|---|---|
| | | μg/m$^3$ | | |
| A | 23 November 2017 | 12 | 8 | 28 |
| | 1 December 2017 | 9 | 17 | 3 |
| | 2 December 2017 | 15 | 19 | 4 |
| B | 10 December 2017 | 52 | 7 | 15 |
| | 11 December 2017 | 37 | 2 | 7 |
| | 12 December 2017 | 62 | 22 | 9 |
| C | 10 December 2017 | 48 | −6 | −1 |
| | 11 December 2017 | 67 | 26 | −4 |
| | 12 December 2017 | 84 | 44 | 13 |
| D | 17 December 2017 | 113 | −1 | −3 |
| | 18 December 2017 | 78 | 23 | 4 |
| | 19 December 2017 | 111 | 24 | 20 |
| E | 27 December 2017 | 85 | 15 | 1 |
| | 28 December 2017 | 107 | 9 | 5 |
| | 29 December 2017 | 30 | −17 | 14 |
| F | 27 March 2018 | 101 | 6 | −2 |
| | 29 March 2018 | 43 | 9 | 14 |
| G | 28 March 2018 | 80 | 45 | 22 |
| | 29 March 2018 | 208 | 60 | 10 |
| Incremental concentration average | | 70.63 | 16.42 | 8.37 |
| Relative incremental concentration | | 42.24% | 19.76% | 16.27% |
| Standard deviation | | 47.08 | 18.59 | 8.98 |

It can be seen from the Table above that there is a certain difference in the emission of building construction dust between construction sites or even between different days of a construction site. Further analysis found that the construction site size, construction process and construction activities and the external environment of each day were different. It is because of these differences that make each construction site unique, which is reflected on the monitoring data as a certain of deviation. The TSP incremental concentration had the highest deviation, the PM$_{10}$ incremental concentration was second, and the PM$_{2.5}$ incremental concentration had the smallest standard deviation. In addition, we can see that PM$_{10}$ and PM$_{2.5}$ incremental concentration had negative values in a few days, which is contrary to the empiricism, and the TSP were not in this kind of situation. This may be due to the small emission intensity of PM$_{10}$ and PM$_{2.5}$ in building construction dust amplifies the monitoring error, which is a systematic error that can be taken into account when calculating the average.

Overall, from the perspective of the absolute value of incremental concentration average of building construction dust, the average daily TSP incremental concentration of the seven construction sites was 70.63 μg/m$^3$, the PM$_{10}$ incremental concentration was 16.42 μg/m$^3$, and the PM$_{2.5}$ incremental concentration was 8.37 μg/m$^3$. The order of incremental concentration average from large to small was TSP, PM$_{10}$, PM$_{2.5}$. Since the TSP contains PM$_{10}$ and PM$_{2.5}$, the result is reasonable in data size. Since the seven construction sites contained various stages of construction, the average level of dust impact from building construction site on surrounding area can be reflected to some extent without considering other factors.

From the perspective of the relative incremental concentration average of building construction dust, compared with the upwind direction concentration, the average daily TSP, $PM_{10}$ and $PM_{2.5}$ concentration increased by 42.24%, 19.76% and 16.27%, respectively. Therefore, the construction sites contributed most to TSP, followed by $PM_{10}$ and $PM_{2.5}$. The above analysis shows that the building construction dust mainly influenced the particle concentration with larger size on surrounding area.

The Qingyuan was in winter and spring between the monitoring period and the climate was relatively dry; thus, the results above maybe higher than the annual average of building construction dust [48].

### 4.4. Particle Size Distribution in Building Construction Dust

This section is to understand the composition of different particle sizes in building construction dust. According to the analysis results of the incremental concentration of the up-downwind direction, the ratio of the incremental concentration of different dust monitoring indicators is calculated. Taking the TSP incremental concentration as 1, the relative of $PM_{10}$ and $PM_{2.5}$ can be calculated, as shown in Figure 5.

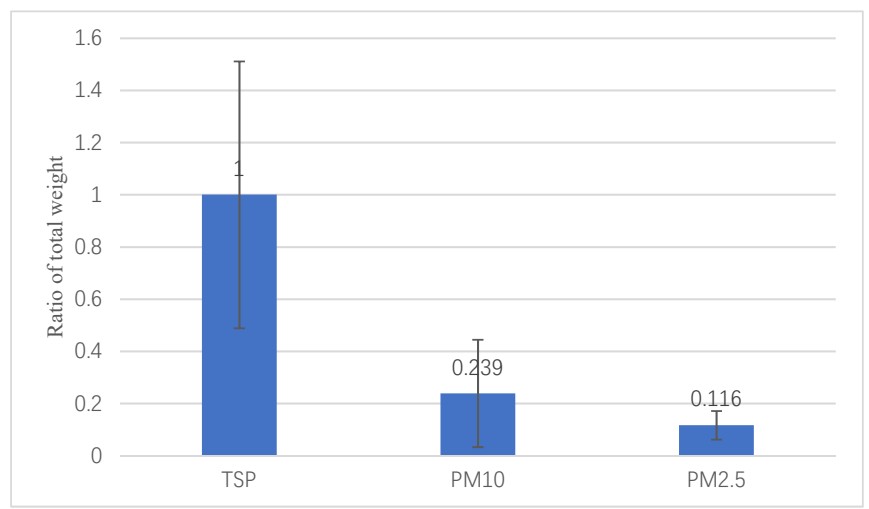

**Figure 5.** Particle size distribution in building construction dust.

It can be seen from Figure 5 that the $PM_{10}$/TSP is 0.239, indicating that about 76.1% of the particulate matter in the building construction dust emission are larger than 10 μm. This part of the particulate matter will naturally settle under gravity within a few hours, and usually can only move within a short distance, which cannot affect the area far from the construction site [49]. The PM2.5/TSP was 0.116, which indicates that about 11.6% of the particulate matter in the construction dust emission was less than 2.5 μm. This part of the particulate matter can be suspended in the air for a long time without precipitation and transported to thousands of kilometers away by wind [50], and it poses a greater risk to human health [51]. Although the proportion of this part of the particulate matter is small, its influence cannot be ignored. The remaining 12.3% of the building construction dust was a particle with a size between 2.5 μm and 10 μm, and its influence range was between the above two kinds of particulate matter. In general, in building construction dust, TSP:$PM_{10}$:$PM_{2.5}$ = 1:0.239:0.116.

### 4.5. Analyzing the Main Influencing Factors of Building Construction Dust

In this section, the Pearson correlation coefficient was adopted to analyze the degree of correlation between building construction dust and other factors to determine whether this factor is the main factor affecting building construction dust emission. The external environment and construction intensity are two important sources of influence. In the field research, it was found that each construction site has adopted similar dust-proof measures, e.g., construction site wall spray, construction vehicles wash

at entrance and exit and sprinkling water on the road inside the site, and met the basic requirements of dust prevention; thus, this article did not analyze this factor.

### 4.5.1. Correlation Analysis of Building Construction Dust and Basic Ambient Air Quality

In this study, the air quality data released by the environmental protection department of Qingyuan was selected as the basic ambient air quality. Since Qingyuan City Environmental Protection Department only publishes the data of $PM_{10}$ and $PM_{2.5}$, this paper only analyzed the correlation between $PM_{10}$ and $PM_{2.5}$ incremental concentration in building construction dust and environmental monitoring points' data. Table 8 shows the results.

**Table 8.** Correlation analysis of building construction dust and basic ambient air quality.

|  |  | **Environmental Monitoring Point $PM_{10}$ Concentration** |
|---|---|---|
| $PM_{10}$ incremental concentration | Pearson Correlation | −0.107 |
|  | significance (two-tailed) | 0.664 |
|  | N | 19 |
|  |  | Environmental monitoring point $PM_{2.5}$ concentration |
| $PM_{2.5}$ incremental concentration | Pearson Correlation | 0.176 |
|  | significance (two-tailed) | 0.470 |
|  | N | 19 |

Analyzing Table 8, it shows that there is no significant correlation between $PM_{10}$ and $PM_{2.5}$ incremental concentration and environmental monitoring points' data, which indicates that the basic ambient air quality will not have a significant impact on building construction dust emission.

### 4.5.2. Correlation Analysis of Building Construction Dust and Meteorological Factors

Through to the monitoring of meteorological data, 19 sets of the meteorological data including wind speed, temperature, humidity and atmospheric pressure of each site were obtained. The meteorological data were calculated by the average of the day. As shown in Figure 6, 19 sets of data were drawn into a line chart. Correlation analysis was carried out between dust incremental concentration and meteorological data. The results are shown in Table 9.

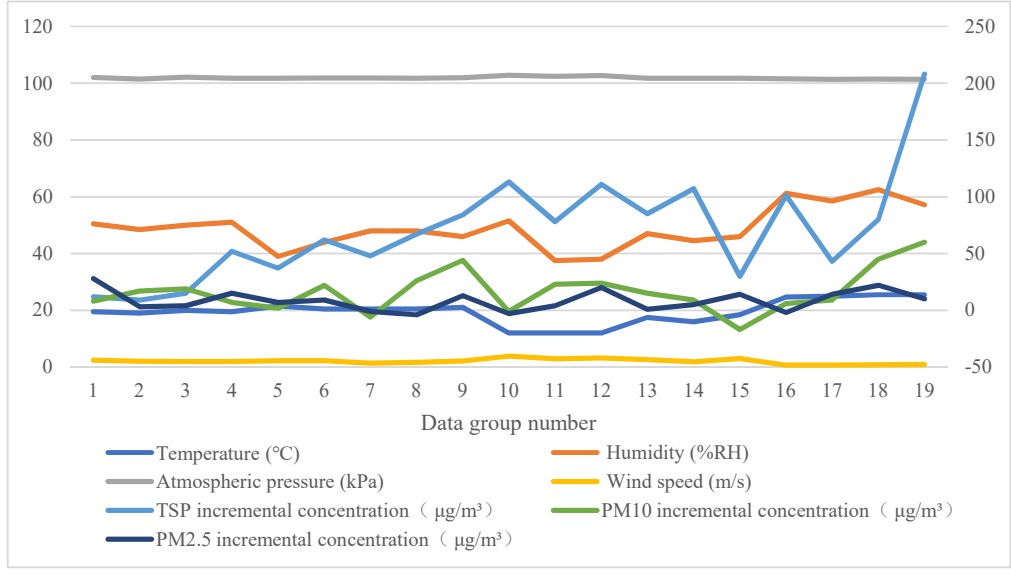

**Figure 6.** Line chart of dust incremental concentration and meteorological data. Note: dust incremental concentration uses the secondary vertical axis.

**Table 9.** Correlation analysis of building construction dust and meteorological factors.

|  |  | Temperature | Humidity | Atmospheric Pressure | Wind Speed |
|---|---|---|---|---|---|
| TSP incremental concentration | Pearson Correlation | 0.004 | 0.178 | 0.054 | −0.105 |
|  | Significance (two-tailed) | 0.986 | 0.465 | 0.827 | 0.669 |
|  | N | 19 | 19 | 19 | 19 |
| $PM_{10}$ incremental concentration | Pearson Correlation | 0.303 | 0.207 | −0.191 | −0.323 |
|  | Significance (two-tailed) | 0.207 | 0.395 | 0.433 | 0.177 |
|  | N | 19 | 19 | 19 | 19 |
| $PM_{2.5}$ incremental concentration | Pearson Correlation | 0.151 | 0.078 | −0.046 | −0.008 |
|  | Significance (two-tailed) | 0.536 | 0.750 | 0.853 | 0.973 |
|  | N | 19 | 19 | 19 | 19 |

Table 9 reveals that the TSP, $PM_{10}$ and $PM_{2.5}$ incremental concentration did not pass the significant test with the meteorological factors, indicating that building construction dust emission was not significantly correlated with any single meteorological factor.

The reasons caused this result may be: (1) building construction dust is affected by many factors. Construction activities are the direct factor that produces building construction dust and have great influence [28] on the building construction dust much more than meteorological factors. (2) In the monitoring period, the meteorological factors did not change too much, while the construction activities on the site are significantly different; thus, it may eliminate the influence of meteorological factors on building construction dust to some extent.

Additionally, although precipitation is a main influencing factor of dust [47], the monitoring equipment was not appropriate for working in the rainy weather and construction activities will be significantly reduced or even suspended in rainy days. Thus, sunny and cloudy days were selected as monitoring time in this study and meteorological factors mentioned above did not include precipitation.

Therefore, it can be considered that building construction dust emission is not significantly correlated with any single meteorological factor when it changes not too much. To some extent, it is consistent with the research conclusions of urban $PM_{10}$ and $PM_{2.5}$ by Ge [52].

### 4.5.3. Correlation Analysis of Building Construction Dust and Construction Intensity

The dust incremental concentration is plotted as a line graph with the daily working hours and daily transport trips, respectively, as shown in Figure 7. The correlation analysis results are shown in Table 10.

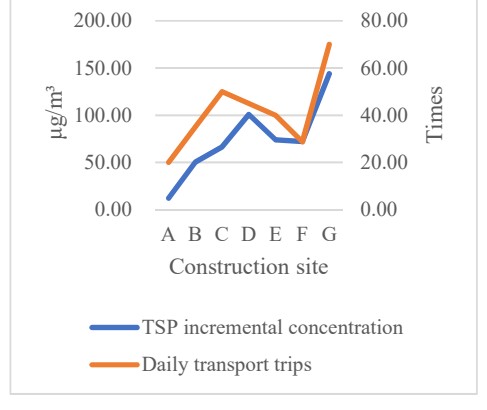 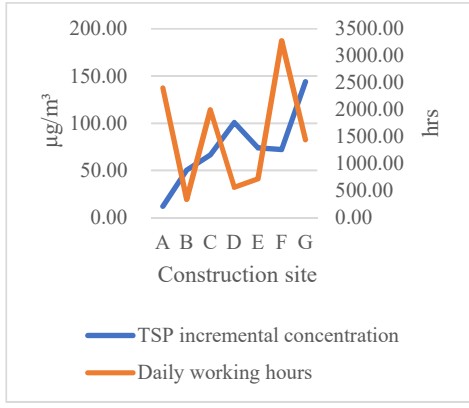

**Figure 7.** *Cont.*

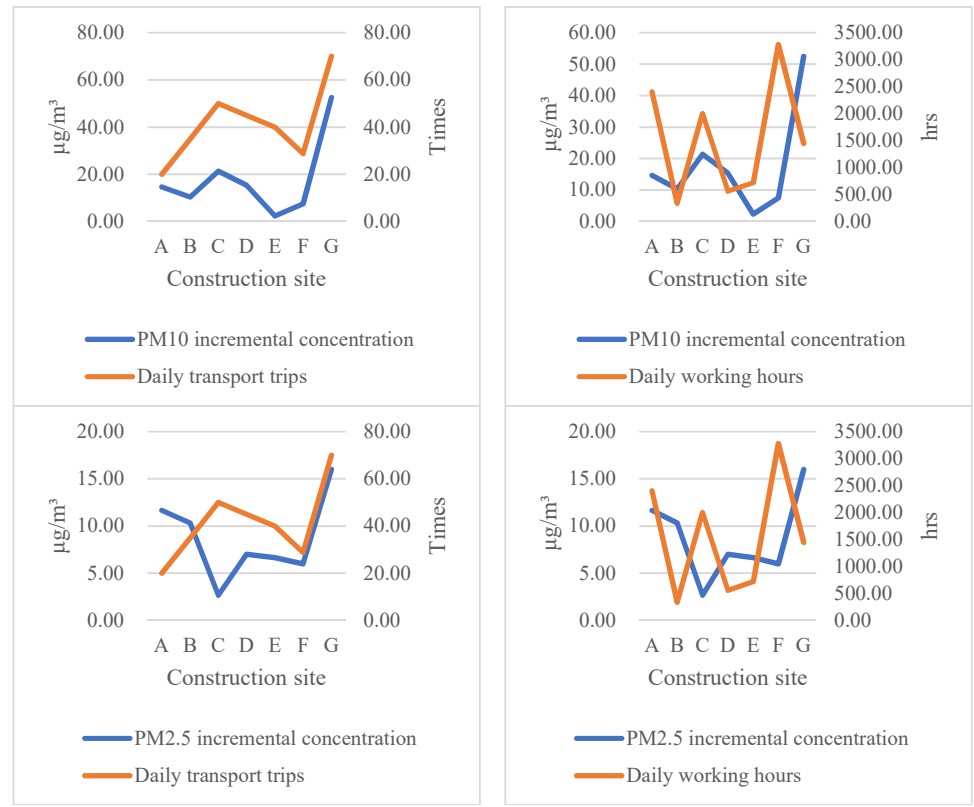

**Figure 7.** Line chart of dust incremental concentration and construction strength intensity.

**Table 10.** Correlation analysis of building construction dust and construction intensity.

|  |  | Daily Working Hours | Daily Transport Trips |
|---|---|---|---|
| TSP incremental concentration | Pearson Correlation | 0.136 | 0.890 ** |
|  | Significance (two-tailed) | 0.771 | 0.007 |
|  | N | 7 | 7 |
| $PM_{10}$ incremental concentration | Pearson Correlation | 0.065 | 0.801 * |
|  | Significance (two-tailed) | 0.890 | 0.030 |
|  | N | 7 | 7 |
| $PM_{2.5}$ incremental concentration | Pearson Correlation | −0.333 | 0.268 |
|  | Significance (two-tailed) | 0.465 | 0.562 |
|  | N | 7 | 7 |

** Correlation is significant at the 0.01 level (two-tailed). * Correlation is significant at the 0.05 level (two-tailed).

It can be seen from Table 10 that there is no correlation between dust incremental concentration and the daily working hours. The TSP and $PM_{10}$ in the building construction dust have a strong positive correlation with the daily transport trips, and the correlation with the TSP incremental concentration is greater than $PM_{10}$ incremental concentration. On the other hand, the correlation between $PM_{2.5}$ incremental concentration and daily transport trips is not obvious. Most of the construction sites are unpaved roads. When construction vehicles are driving in the venue, the road dust will rise and generate a lot of dust. According to previous study, more than 80% of the particulate matter by this is larger than 10 μm [53]. It is consistent with the results in the previous analysis, that the large-diameter particulate matter is generated by the building construction dust. Construction trips are one of the main influencing factors of building construction dust, while the daily working hours as construction intensity did not have significant correlation.

## 5. Conclusions

In this paper, the monitoring method of up-downwind direction was adopted. The monitoring points were distributed at the boundary of the construction site to monitor the dust impacts on the surrounding areas. Comprehensive monitoring indicators including TSP, $PM_{10}$ and $PM_{2.5}$ were selected. Thus, dust impact from building construction site on surrounding area can be quantified and characterized comprehensively. At the same time, the data of meteorological and construction intensity were collected to determine the main factors affecting the construction dust emission, which can provide a basis for reducing the impact of dust generated by construction activities on the surrounding area. The main conclusions of the article are as follows:

Through on-site monitoring of seven construction sites in Qingyuan City, this study found that the dust emission level of construction activities is relatively high. The average daily TSP incremental concentration was 70.63 $\mu g/m^3$, the $PM_{10}$ incremental concentration was 16.42 $\mu g/m^3$ and the $PM_{2.5}$ incremental concentration was 8.37 $\mu g/m^3$. In addition, compared with the upwind direction concentration, the construction site makes downwind direction TSP, $PM_{10}$ and $PM_{2.5}$ concentration increased by 42.24%, 19.76% and 16.27%, respectively, which indicates that the construction activity had a significant impact on the air quality around the construction surrounding areas. At the same time, in building construction dust, TSP:$PM_{10}$:$PM_{2.5}$ = 1:0.239:0.116, which is mainly large diameter particulate matter. This part of the particulate matter will quickly settle under gravity and will only have an effect in the smaller surrounding area. Moreover, according to the particle concentration limit table, it was found that the over-standard rate of different monitoring indicators for the same construction site had remarkable difference. This shows that using different indicators had a significant impact on the evaluation of the impact of construction sites on air quality. Comprehensive monitoring indicators can be a proper strategy to evaluate air quality in different aspects.

Regarding the main factors affecting the building construction dust emission, the results show that building construction dust emission was not significantly correlated with any single meteorological factor when it did not change too much. Building construction dust emission was also not significantly correlated with daily working hours. Construction vehicles were one of the main influencing factors of building construction dust. Therefore, dust-proof measures, such as cleaning the in and out of the vehicle and sprinkling water on the construction road, are particularly necessary.

The research findings show that the construction sites have an important impact on the dust concentration in the surrounding area of the downwind direction. At the same time, the main factors affecting the construction dust emission were explored, which provides a theoretical basis for the treatment of building construction dust. Future research can be carried out taking into account the following aspects: (1) highlighting the differences in emissions from building construction dust at different construction stages to carry out targeted treatment; (2) studying the attenuation of the downwind dust concentration and accurately assessing the sphere of the impact of building construction dust; (3) evaluating the effect of using dust-proof measures to suppress dust emissions from building construction; (4) developing a new formula of dust incremental concentration to assess the dust impact on surrounding area by more data.

**Author Contributions:** Conceptualization, H.Y. and G.D.; methodology, H.Y.; software, G.D. and K.F.; validation, H.Y., G.D. and K.F.; formal analysis, H.Y. and G.D.; investigation, G.D., H.L., Y.W. and L.Z.; resources, H.Y. and Y.W.; data curation, G.D.; writing—original draft preparation, G.D.; writing—review and editing, K.F. and H.L.; visualization, G.D.; supervision, H.Y. and Q.S.; project administration, H.Y.; funding acquisition, H.Y.

**Funding:** This research was funded by the National Natural Science Foundation of China and Guangzhou Federation of Social Sciences, grant number 71403090, 71501074, 2016GZYB63, 15Q06.

**Acknowledgments:** The authors wish to express their sincere gratitude to the National Natural Science Foundation of China and Guangzhou Federation of Social Sciences, for the generous funding support to the projects (71403090, 71501074, 2016GZYB63, 15Q06) on which this paper is based.

**Conflicts of Interest:** The authors declare no conflict of interest.

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
