# Peer review of "Field Evaluation of the Dust Impacts from Construction Sites on Surrounding Areas: A City Case Study in China"

_sustainability, doi:10.3390/su11071906_

Reviewer 1 Report

Construction activities and the corresponding characteristics of construction dust is studied here for finding the way of limiting its effects. To reduce the influence of surrounding areas, seven representative construction sites in Qingyuan city, China were investigated as empirical cases for field evaluation. In the experiment, up-downwind method was adopted to monitor and collect TSP, PM10 and PM2.5 concentrations, meteorological data and construction activities of each site for 2 to 3 days and 18 hours in each day. The results show that the construction site originate in the surrounding areas a concentration increase of average daily TSP, PM10 and PM2.5 by 42.24%, 19.76% and 16.27%, respectively. The proportion of TSP, PM10 and PM2.5 is 1, 0.239 and 0.116, respectively, in building construction dust. The big particulate matter is the major constituent and the distance of its influence is low. In addition, construction vehicles are one of the main influencing factors for building dust. It is found that building construction dust is not significantly correlated with any single meteorological factor.

General comments

The statement that building construction dust is not significantly connected with any meteorological influence is surprising because in the literature review precipitation and wind were found as significant factors. This discussion should be included whereas the details are given in the chapter Results.

The paper addresses relevant scientific questions. The paper presents novel concepts, ideas and tools.

The scientific methods and assumptions are valid and clearly outlined so that substantial conclusions are reached.

The description of experiments and calculations are sufficiently complete and precise to allow their reproduction by fellow scientists.

The quality, information and caption of the figures are good.

The related work is well cited.

Title and abstract reflect the whole content of the paper.

The overall presentation is well structured and clear. The language is fluent but can be improved in some parts.

The abbreviations and units are generally correctly defined and used.

Specific Comments

The figure and table captions should be understandable by themselves – e.g. abbreviations should be explained and given values should be defined.

In Fig. 6 a second axis should be used for TSP concentration so that the variation of the other parameters is more visible.

Technical corrections

The references are written very different but should follow the journal guidelines.

Author Response

Reviewer #1:

We would like to thank the reviewer #1 for providing feedbacks which enabled us to identify and improve further the weaknesses of our paper. Below you can find the review comments and the responses to these comments. In the revised manuscript, we have also highlighted the changes that were made.

1. The statement that building construction dust is not significantly connected with any meteorological influence is surprising because in the literature review precipitation and wind were found as significant factors. This discussion should be included whereas the details are given in the chapter Results.

Respond

Indeed, the statement of this conclusion is not clearly.

The discussion is added in Part 4.5.2: “The reasons caused this result may be: (1) Building construction dust is affected by many factors. Construction activities are the direct factor that produces building construction dust and have great influence [28] on the building construction dust much more than meteorological factors. (2) In the monitoring period, the meteorological factors do not change too much, while the construction activities on the site are significantly different, so it may eliminate the influence of meteorological factors on building construction dust to some extent. Additionally, although precipitation is a main influencing factor of dust [47], the monitoring equipment is not appropriate for working in the rainy weather and construction activities will be significantly reduced or even suspended in rainy days. Thus, sunny and cloudy days were selected as monitoring time in this study and meteorological factors mentioned above are not included precipitation.”

The statement is revised to “it can be considered that building construction dust emission is not significantly correlated with any single meteorological factor when it changes not too much.”

At the same time, the abstract and conclusion are revised accordingly.

2. The paper addresses relevant scientific questions. The paper presents novel concepts, ideas and tools.

The scientific methods and assumptions are valid and clearly outlined so that substantial conclusions are reached.

The description of experiments and calculations are sufficiently complete and precise to allow their reproduction by fellow scientists.

The quality, information and caption of the figures are good.

The related work is well cited.

Title and abstract reflect the whole content of the paper.

The overall presentation is well structured and clear. The language is fluent but can be improved in some parts.

The abbreviations and units are generally correctly defined and used.

Respond

Thanks very much for the recognition of the paper.

3. The figure and table captions should be understandable by themselves – e.g. abbreviations should be explained and given values should be defined.

In Fig. 6 a second axis should be used for TSP concentration so that the variation of the other parameters is more visible.

Respond

We have checked all figures and tables and explain the abbreviations and define the given values.

In figure 3, the name of the construction site is added.

In figure 5, the definition of the vertical axis is added.

In figure 6, a second axis is used for TSP, PM10 and PM2.5 incremental concentration, and a note is added under the figure.

4. The references are written very different but should follow the journal guidelines.

Respond

Format of references are checked to make them follow the journal guidelines.

Reviewer 2 Report

I think that this paper is a valuable information on the environmental assessment to estimate the impact of the building construction on air quality. I expect you to proceed this study and to develop a new formula of the dust emission intensity by the construction trip, using the observed data of dust concentration and wind velocity.

Author Response

Reviewer #2:

We would like to thank the reviewer #2 for providing feedbacks which enabled us to identify and improve further the weaknesses of our paper. Below you can find the review comments and the responses to these comments. In the revised manuscript, we have also highlighted the changes that were made.

1. Will you revise this sentence.

“Followed is the internal activities of construction sites.”

Response

This sentence may not be very clear, so it is revised to “The following is the internal activities of construction sites which are also one of the main influencing factors of building construction dust.”.

2. Will you use more physical word, such as less than ? micron meter diameter.

Response

This sentence is revised to “Peaks of particulate matter (including PM10, PM2.5 and PM0.1) during drilling and cutting activities were 4 and 14 times higher than the background value.”.

3. You need to erase one "This".

You had better to use semi-Colon ";", not ":".

You need to add "," after this word.

Response

Thanks for pointing out the formatting errors. All errors have been revised in the manuscript.

4. Will you add more information on this 2050, such as manufacture name.              

Response

We are sorry we didn't write the instrument information clearly. More information is added: “air/smart integrated sampler (2050)”.

5. Will you enlarge the characters of this maps, and add the name of the construction site.

Response

The characters of maps are enlarged and the name of the construction site is added in figure 3.

6. You should not to say this incremental concentration average to be a dust emission, because the unit of dust emission is ? microgram/m2/sec. You had better to say this number as incremental concentration average. It may be difficult to evaluate the dust emission intensity from the observed data. If you hope to evaluate the dust emission intensity, the value of concentration divided by wind velocity (microgram/m3  divided m/s) correspond with it.

Response

The reviewer #2 pointed out that “dust emission” is not suitable in this sentence and we agree it.

Therefore, we revise it to “incremental concentration average”. In addition, we also check the whole paper and revise some of the statements about “dust emission” that are not suitable. Please see the revised paper for details.

7. You had better to add the definition of vertical axis as ratio of total weight. Generally speaking, the vertical axis of particle size distribution is shown as total particle number of the particle size.

Response

We add the definition of vertical axis as the ratio of total weight in figure 5.

We agree that in general, particle size distribution is shown as the total particle number of the particle size. However, due to the limitations of monitoring methods, the mass concentration of particulate matter is monitored in this paper. We can only use the ratio of mass concentration of particulate matter to approximately represent the particle size distribution.

8. Will you explain what kind of dust proof measures are adapted in each site, briefly.

Response

Several main dust proof measures are added in Part 4.5: “e.g. construction site wall spray, construction vehicles wash at entrance and exit and sprinkling water on the road inside the site”.

9. Will you explain more in detail about the definition of the vertical axis and its unit.

Response

Since the figure contains information about a variety of data with different unit, the numbers on the vertical axis only represent values and do not contain units. The units of each type of data are given in the legend. In order to make the figure more clearly, the TSP, PM10 and PM2.5 increment concentration are changed to use the secondary vertical axis, as is shown in figure 6.

10. I think that this paper is a valuable information on the environmental assessment to estimate the impact of the building construction on air quality. I expect you to proceed this study and to develop a new formula of the dust emission intensity by the construction trip, using the observed data of dust concentration and wind velocity.

Response

Thanks for the recognition of the paper. We seriously consider the research direction you pointed out and agree it is a valuable attempt. However, the data required for a reliable formula may be larger than the observed data. Also, many other factors should be comprehensively considered for a reliable formula. Therefore, we hope to make further research in the field of building construction dust.

We have added the suggestion of future research in Part “Conclusions”: “(4) develop a new formula of dust incremental concentration to assess the dust impact on surrounding area by more data”.

Round  2

Reviewer 1 Report

Acceptance of this paper according to the revisions of the paper